# Sex Differences in Gut Microbiota and Their Relation to Arterial Stiffness (MIVAS Study)

**DOI:** 10.3390/nu17010053

**Published:** 2024-12-27

**Authors:** Rita Salvado, Cristina Lugones-Sánchez, Sandra Santos-Minguez, Susana González-Sánchez, José A. Quesada, Rocío Benito, Emiliano Rodríguez-Sánchez, Manuel A. Gómez-Marcos, Pedro Guimarães-Cunha, Jesús M. Hernandez-Rivas, Alex Mira, Luis García-Ortiz

**Affiliations:** 1Primary Care Research Unit of Salamanca (APISAL), Salamanca Primary Healthcare Management, Castilla y León Regional Health Authority (SACyL), Institute of Biomedical Research of Salamanca (IBSAL), 37007 Salamanca, Spain; crislugsa@gmail.com (C.L.-S.); gongar04@gmail.com (S.G.-S.); magomez@usal.es (M.A.G.-M.); lgarciao@usal.es (L.G.-O.); 2Research Network on Chronicity, Primary Care and Health Promotion (RICAPPS), 37005 Salamanca, Spain; 3Cancer Research Centre, Institute of Molecular and Cellular Biology of Cancer (IBMCC), Institute of Biomedical Research of Salamanca (IBSAL), University of Salamanca—CSIC, 37007 Salamanca, Spain; ssantosminguez@gmail.com (S.S.-M.); jmhr@usal.es (J.M.H.-R.); 4Clinical Medicine Department, Miguel Hernandez University, 03550 Alicante, Spain; jquesada@umh.es; 5Network of Research on Chronicity, Primary Care and Health Promotion (RICAPPS), 03550 Alicante, Spain; 6Department of Medicine, University of Salamanca (USAL), 37007 Salamanca, Spain; 7Unidade Local de Saúde do Alto Ave, Center for the Research and Treatment of Arterial Hypertension and Cardiovascular Risk, 4835-044 Guimarâes, Portugal; 8Life and Health Sciences Research Institute (IICVS), School of Medicine, Minho University, 4704-553 Braga, Portugal; 9Haematology Department, Institute of Biomedical Research of Salamanca (IBSAL), University Hospital of Salamanca (USAL), 37007 Salamanca, Spain; 10Department of Health and Genomics, FISABIO Foundation, 46020 Valencia, Spain; alex.mira@fisabio.es; 11CIBER Center for Epidemiology and Public Health, 28029 Madrid, Spain; 12Department of Biomedical and Diagnostic Sciences, University of Salamanca, 37007 Salamanca, Spain

**Keywords:** gastrointestinal microbiome, sex, vascular stiffness, pulse wave velocity, cardio–ankle vascular index, large arteries, short-chain fatty acids

## Abstract

Background: Recent research highlights the potential role of sex-specific variations in cardiovascular disease. The gut microbiome has been shown to differ between the sexes in patients with cardiovascular risk factors. Objectives: The main objective of this study is to analyze the differences between women and men in the relationship between gut microbiota and measures of arterial stiffness. Methods: We conducted a cross-sectional study in Spain, selecting 180 subjects (122 women, 58 men) aged between 45 and 74. Subjects with arterial stiffness were identified by the presence of at least one of the following: carotid–femoral pulse wave velocity (cf-PWV) above 12 mm/s, cardio–ankle vascular index (CAVI) above nine, or brachial–ankle pulse wave velocity (ba-PWV) above 17.5 m/s. All other cases were considered subjects without arterial stiffness. The composition of the gut microbiome in fecal samples was determined by 16S rRNA sequencing. Results: We found that women have a more diverse microbiome than men (Shannon, *p* < 0.05). There is also a significant difference in gut microbiota composition between sexes (Bray–Curtis, *p* < 0.01). *Dorea, Roseburia*, and *Agathobacter,* all of them short-chain fatty-acid producers, were more abundant in women’s microbiota (log values > 1, *p*-value and FDR < 0.05). Additionally, *Blautia* was more abundant in women when only the subjects with arterial stiffness were considered. According to logistic regression, *Roseburia* was negatively associated with arterial stiffness in men, while *Bifidobacterium* and *Subdoligranulum* were positively related to arterial stiffness. Conclusions: In the Spanish population under study, women had higher microbiome diversity and potentially protective genera. The host’s gender determines the influence of the same bacteria on arterial stiffness. Trial Registration Number: NCT03900338.

## 1. Introduction

Cardiovascular disease (CVD) remains a leading cause of morbidity and mortality worldwide [1], and growing evidence suggests significant sex differences in its prevalence, progression, and outcomes. In Spain, data from 2022 indicated that circulatory diseases, including coronary disease, cerebrovascular disease, cardiac insufficiency, and hypertensive disease, accounted for 267.87 deaths per 100,000 men and 174.12 deaths per 100,000 women [2].

Recent efforts have concentrated on enhancing the identification of individuals at high risk for cardiovascular events by employing biomarkers that signal early changes before overt disease manifests [3]. Arterial stiffness has proven to be a reliable predictor of cardiovascular disease across diverse populations [4], including the general population, hypertensive individuals, the elderly, and patients with type 2 diabetes or end-stage renal disease [5]. The main determinants of arterial stiffness encompass major cardiovascular risk factors [1], such as age, hypertension, gender, obesity, smoking, dyslipidemia, hypothyroidism, and influences from genetic predisposition, systemic inflammation, and gut microbiota composition.

The transition from a healthy microbiota to dysbiosis involves increased pathobionts and is often triggered by environmental factors [6]. While age and genetic composition play significant roles, diet and lifestyle are key contributors to microbiome variation. The Mediterranean diet enhances microbial diversity and promotes beneficial bacteria [7], whereas artificial sweeteners and dietary emulsifiers can disrupt gut microbiota and cause inflammation [8]. Variations also exist between omnivores and vegans [9] and between those with high protein intake and those with high carbohydrate intake [10]. Obesity is linked to reduced fecal microbial diversity [11]. Antibiotic use can cause rapid declines in microbial diversity [12].

The gut microbiome is also being studied for its differences between males and females [13], and these differences have been recently documented in the Spanish population [14]. In this study involving 530 Spanish patients, the gut microbiome was primarily characterized by *Firmicutes* (~53.9%) and *Bacteroidota* (~37.2%). The most prevalent genera included *Bacteroides* (~18.4%) and *Faecalibacterium* (~12.5%), followed by *Prevotella* (6.7%), Alistipes (~3.4%), and *Oscillospiraceae* (~2.3%). The gut microbiota were rich (Chao 283) and diverse (Shannon 4.3). The levels of the phylum Proteobacteria and the genus *Faecalibacterium* were significantly greater in males compared to females. 

Recent research has highlighted the potential role of the gut microbiome in cardiovascular sex-specific variations, particularly concerning arterial stiffness, a key indicator of cardiovascular health [15,16,17]. In a 2018 study, Menni et al. found evidence linking gut microbial diversity to arterial stiffness in middle-aged women [15]. The research showed an inverse relationship between gut microbial diversity and pulse wave velocity (PWV), a measure of arterial stiffness. This association remained significant after adjusting for confounding factors, indicating an independent link between gut microbiome composition and vascular health specific to women. In a recent study, Cuadrat et al. explored the connection between human gut microbiota and vascular stiffness [18], identifying particular microbial taxa and pathways associated with arterial stiffness. Their findings underscored the role of gut microbiota in vascular health and shed light on the potential mechanisms linking microbial communities to cardiovascular risk.

Furthermore, there is evidence of a connection between gut microbiota and the risk factors for and end-organ damage of cardiovascular disease. Virwani et al. found that gut microbiota dysregulation was strongly linked to hypertension in women but not in men based on 24 h ambulatory blood-pressure monitoring [19]. Garcia-Fernandez et al. reported that the dysbiosis of the intestinal microbiota associated with coronary heart disease appears to be partially sex-specific [20]. 

The findings emphasize the need to consider sex-specific mechanisms when developing strategies and therapies to reduce cardiovascular disease incidence and recurrence, particularly those targeting arterial stiffness. This underscores the need for more comprehensive studies focusing on sex-specific gut microbiome profiles and their connection to measures of arterial stiffness and overall cardiovascular health.

Considering these observations, the main objective of our study was to investigate sex-specific differences in the composition and function of the intestinal microbiota in a Spanish population free from cardiovascular disease and analyze the influence of sex on the relationship between the intestinal microbiome and arterial stiffness.

## 2. Material and Methods

### 2.1. Design and Setting

The MIVAS study was conceived as an observational, multicentric, case-control investigation. The first phase (a cross-sectional study) was developed in the Primary Care Research Unit of Salamanca (APISAL), which belongs to the Biomedical Research Institute of Salamanca (IBSAL) [21]. The Cancer Research Center of Salamanca and the University of Valencia’s Department of Health and Genomics analyzed microbiota. The study has been registered at ClinicalTrials.gov, with the identifier NCT03900338 (https://clinicaltrials.gov/ct2/show/NCT03900338 (accessed on 25 December 2024)). The Strobe checklist is included in the Appendix A.

### 2.2. Study Population

The MIVAS study involved individuals aged 45 to 74 who are free from cardiovascular disease. This article analyzes the initial 180 subjects enrolled in the study.

The study’s subjects were recruited from two sources: 8% from an EVA study database [22] and 92% from Salamanca primary care centers (Appendix A). Regardless of where they were recruited, all subjects underwent the same evaluation procedures detailed in the section on variables and measuring instruments.

#### 2.2.1. Selection Criteria

Inclusion criteria: patients between 45 and 74 years old who agreed to participate in the study.

Exclusion criteria: history of CVD (ischemic heart disease or stroke, peripheral arterial disease or heart failure), diabetes mellitus, renal failure in terminal stages (glomerular filtration rate below 30%), chronic inflammatory diseases, inflammatory bowel disease, body mass index >40 kg/m^2^, oncologic disease diagnosed in the last five years and/or under treatment, terminal conditions, antibiotic use within the previous 15 days, and those who refused to sign the informed consent.

#### 2.2.2. Sample Size and Power Calculation

With the 180 subjects included in the MIVAS I study, accepting a risk alpha of 0.05 and a common standard deviation in the Shannon index at the genus level of 0.25, in a bilateral contrast with 122 subjects in the first group (females) and 58 in the second (males), the statistical power is 86% to detect as statistically significant a mean difference of 0.12 units; this implies an effect size of 0.5.

#### 2.2.3. Patient and Public Involvement

The subjects were not involved in the study’s conception but helped with recruitment through their organizations. After completing the clinical evaluation, each participant was sent a detailed report and suggestions for improving their lifestyle.

### 2.3. Variables and Measurement Instruments

A standardized protocol was used to train two research nurses to take measurements and answer questionnaires before the study began [21].

#### 2.3.1. Sociodemographic and Clinical Variables

The following data were collected when the study was initiated: age, sex, hypothyroidism, and drug consumption. Hypertension and dyslipidemia were evaluated and classified following the most recent guidelines from the European Society of Cardiology [23,24].

#### 2.3.2. Anthropometric Measurements

Office or clinical blood pressure (BP) was found after three measurements of systolic BP (SBP) and diastolic BP (DBP), using the average of the last two readings. Measures were performed using a validated OMRON model, following the European Society of Hypertension’s (ESH) [25] recommendations. Body mass index (BMI) was calculated as weight (kg) divided by height (m) squared [21,25].

#### 2.3.3. Habits and Lifestyles

We used the food questionnaire from the PREDIMED study group, the Mediterranean Diet Adherence Screener (MEDAS) [26], to assess adherence to the Mediterranean diet. This query has 14 questions, each scored as 0 or 1. Mediterranean-diet compliance was assumed when the total score was ≥9 points. Tobacco and alcohol consumption were evaluated using standardized questionnaires [21].

#### 2.3.4. Laboratory Measurements

Venous blood samples were taken between 08:00 and 09:00 h, after 12 h of fasting. Automated enzymatic methods were used to measure hemogram, plasma glucose, glycated hemoglobin, creatinine, uric acid, liver function, lipids, and thyroid function.

#### 2.3.5. Gut Microbiota Measurements

Participants provided and collected stool samples with the OMNIgene GUT (OMR-200) kit (DNAgenoteck, Ottawa, ON, Canada), which allows for storing stabilized DNA at room temperature for 60 days. The Cancer Research Center in Salamanca processed specimens, and DNA quality and amount were measured using techniques including TapeStation 4200 (Agilent, Santa Clara, CA, USA), Qubit 4.0 fluorometer, and Nanodrop (Thermo Fisher Scientific, Waltham, MA, USA) [21].

#### 2.3.6. Amplicon and Illumina Sequencing of Bacterial 16S rRNA Genes

The V5–V6 region of the 16S rRNA gene was amplified using the following primers with the sequence [27]: V5F_Nextera 5′-RGGATTAGATACCC-3′ and V6R_Nextera 5′-CGACRRCCATGCANCACCT-3′; and added Illumina adapter overhang nucleotide sequences: ü Forward overhang: 5′ TCGTCGGCAGCGTCAGATGTGTATAAGAGACAG-[locus-specific sequence]; ü Reverse overhang: 5′ GTCTCGTGGGCTCGGAGATGTGTATAAGAGACAG-[locus-specific sequence].

To analyze the microbial communities, we amplified the V5–V6 regions of the 16S rRNA gene with a primer set, purified the resulting amplicons, and added indexes in a second PCR reaction. Each sample was designated using a unique combination of two indexes (S5XX-N7XX). We pooled the final amplicon libraries equimolarly and sequenced them on the Illumina MiSeq platform (Illumina, Inc. San Diego, CA, USA) using a v3 (600-cycle) reagent kit. We processed the raw sequence data using our own pipeline and grouped quality passing-filter readings into operational taxonomic units (OTUs). As part of our quality control measures, we discarded paired ends with an overlap of less than 200 nt. We removed chimeric sequences using de novo chimera detection in USEARCH (https://www.drive5.com/usearch/; full access 25 December 2024) on a per-sample basis [28].

#### 2.3.7. Vascular Function Assessment

Carotid–femoral pulse wave velocity (cf-PWV) and Central Augmentation Index (CAIx). The SphygmoCor System (AtCor Medical Pty Ltd., Head Office, West Ryde, Australia) was utilized to estimate these parameters. It uses a mathematical transformation to calculate aortic pulse wave from pulse wave analysis performed in the radial artery, with the patient sitting and resting their arm on a rigid surface. CAIx and central blood pressure were estimated from aortic wave morphology. The cf-PWV was calculated by comparing the carotid–femoral artery pulse wave (patient in a supine position) delay with the ECG wave and dividing it by the distance from the sternal notch to the carotid and femoral arteries at the sensor location. An abnormal cf-PWV reading was determined when the value exceeded twelve m/s [29].

Cardio–ankle vascular index (CAVI) and brachial–ankle PWV (ba-PWV). The Vasera device VS-2000 (Fukuda Denshi Co., Ltd., Tokyo, Japan) was used to estimate these parameters. The CAVI values and ba-PWV were calculated using specific equations. The CAVI values were classified as usual (CAVI < 8), borderline (8 ≤ CAVI < 9), or abnormal (CAVI ≥ 9), while a ba-PWV greater than 17.5 m/s was considered abnormal [30].

#### 2.3.8. Arterial Stiffness Definition

Arterial stiffness subjects were defined by the presence of at least one of the following: carotid–femoral pulse wave velocity (cf-PWV) above 12 mm/s, cardio–ankle vascular index (CAVI) above 9, or brachial–ankle pulse wave velocity (ba-PWV) above 17.5 m/s. All other cases were considered subjects without arterial stiffness.

### 2.4. Statistical Analysis

The data were recorded using REDCap (Research Electronic Data Capture) https://giap.usal.es/redcap/; full access 25 December 2024) [30]. An X² test was utilized to compare qualitative variables, while for quantitative variables with two categories, a Student’s *t*-test or Mann–Whitney U-test was used depending on the situation.

Microbiome analysis: Microbiome data were filtered to include only features with more than four counts in more than 80% of their values. After standardization, all the studies were performed comparing women with men in all the samples and only in the arterial stiffness group.

Overall, the microbiota structure was assessed by studying phylum abundance. Alpha diversity measures within a single sample [31] were determined using the Shannon–Wiener index [32], and inter-individual Bray–Curtis values (beta diversity) were used to evaluate differences between samples. Principal Coordinates Analysis (PCoA), also known as Metric Multidimensional Scaling (MDS), was used to visualize the pairwise distances between samples using the Bray–Curtis distance.

In the logistic regression analysis, we used the 12 most abundant genera to analyze the microbiome’s relationship with arterial stiffness. Negative binomial models were adjusted to examine the association between the abundance of each genus separately, taking as the dependent variable having or not arterial stiffness. For each genus, an adjustment for age and sex was performed (Model 1), as well as an adjustment for age, sex, body mass index, hemoglobin A1c, total cholesterol, HDL cholesterol, LDL cholesterol, and systolic blood pressure (Model 2), and an adjustment that added Mediterranean diet score was added to Model 3. Odds Ratios (ORs) and 95% confidence intervals (95% CIs) were estimated. Due to the high variability of scales, the abundance was standardized by centering on the mean and dividing by the standard deviation to compare the ORs of each genus.

Alpha and beta diversity analysis was performed using the vegan package (v. 2.5-7) [33] and plotted employing the phyloseq package version 1.40.0 [34]. Differentially abundant microbes were analyzed using EdgeR 4.0 [35] in R. This involved data normalization using the trimmed mean of M-values (TMM), dispersion estimation with empirical Bayes methods, fitting a negative binomial generalized linear model for each genus, and conducting statistical testing with adjustments for multiple testing using methods like Benjamini–Hochberg to control the false discovery rate (FDR). We considered results significant when LogFC was greater than 1 or less than −1 and the false discovery rate (FDR) was less than 0.05. SPSS V.25.0 statistical package (SPSS Inc., Chicago, IL, USA) was used for database management. All the other analyses were conducted in R version 4.2.0. *p* values were corrected using the Benjamini-Hochberg procedure (FDR). *p* values and FDR cutoff values for significance were established at <0.05.

### 2.5. Ethical Considerations

The study was approved by the Committee on Ethics of Research with Medicines in the Health Area of Salamanca on 14 December 2018 (ref: 2018-11-136) and the Ethics Committee for Health of Guimaraes (Portugal) on 15 October 2019 (ref: 67/2019). All participants consented to the Helsinki Declaration, and their data were kept confidential. Regulation (EU) 2016/679 of the European Parliament and of the Council of 27 April 2016 guaranteed the confidentiality of participants’ data.

## 3. Results

We analyzed the demographic and clinical characteristics of the 180 participants (Table 1). Females were 67.8% of the sample. The mean ± SD age of the participants is 62.6 ± 7.5 years, without differences between sexes. SBP (122.1 ± 17.5 vs. 129.8 ± 16.8 mmHg), DBP (77.8 ± 9.9 vs. 82.3 ± 8.6 mmHg), diagnosed hypertension (34.1% vs. 53.4%), and total cholesterol were higher in men (*p* < 0.01 for all). Women had a higher heart rate (69.4 ± 9.8 vs. 63.0 ± 9.7) and HDL cholesterol levels (63.5 ± 14.4 vs. 53.1 ± 11.5 mg/dL) (*p* < 0.01).

Cf-PWV was higher in men (11.2 ± 2.6 m/s) than in women (10.4 ± 2.1 m/s), *p* = 0.04. There were no differences between sexes in ba-PWV (14.2 ± 2.5 m/s) or CAVI (8.4 ± 1.0). The percentage of subjects with arterial stiffness was 38.8%, with no difference by sex Appendix A.

The analysis of phylum relative abundance showed a higher abundance of *Bacteriodota* in men and *Firmicutes* in women (*p* < 0.01) (Figure 1).

An analysis of bacterial communities at the genus taxonomic level showed similar richness (*p* = 0.71), measured by Chao in women (median 82, IQR: 41–107) and men (median 82, IQR: 29–108). We found differences between women and men in alpha diversity (Shannon index: 3.15 ± 0.25 (female) vs. 3.08 ± 0.25 (male); *p* = 0.048). Principal Coordinate Analysis using Bray–Curtis Dissimilarity at the genus level showed differences between the sexes (*p* < 0.01) (Figure 2). No significant differences were observed in alpha or beta diversity measures between subjects with and without arterial stiffness for either sex (women: Shannon, *p* = 0.88, Bray–Curtis, *p* = 0.32; men: Shannon, *p* = 0.94, Bray–Curtis, *p* = 0.81).

When analyzing genus differential expressions between sexes, we found that *Dorea*, *Roseburia*, and *Agathobacter* are more abundant in women’s than in men’s microbiota (log values > 1, *p*-value and FDR < 0.05) Appendix A. When analyzing only people with arterial stiffness beyond those referred, *Blautia* was also more abundant in women (Figure 3 and Appendix A).

In the logistic regression, when considering all the subjects, we found a positive association of *Agathobacter*, *Anaerostipes*, and *Roseburia* with arterial stiffness that was lost after adjusting for clinical risk factors and diet (Figure 4 and Appendix A.

When analyzing women, none of the genera was associated with arterial stiffness in any model (Figure 5 and Appendix A. When examining men, we found that the *Rosuberia* genus was negatively related to arterial stiffness, and *Bifidobacterium* and *Subdoligranulum* were positively associated in Models 2 and 3 (Figure 5 and Appendix A.

## 4. Discussion

The main results show that women had a more diverse gut microbiome than men, with differences in gut microbiota composition at both the phylum and genus levels. *Agathobacter*, *Dorea*, *Roseburia* (all samples), and *Blautia* (only in the arterial stiffness group) were more abundant in women. Logistic regression showed that, in men, *Roseburia* was negatively associated with arterial stiffness, while *Bifidobacterium* and *Subdoligranulum* were positively related to arterial stiffness. No significant associations were found between specific genera and arterial stiffness in women.

Our research shows that women have higher gut microbiome diversity, as indicated by higher Shannon diversity indexes and beta diversity differences between women and men. Similar results were found in large-scale studies of healthy individuals from Belgium, the Netherlands [36], Ukraine [13], and Japan [37]. Our observation of higher Firmicutes in women and higher *Bacteroidota* in men aligns with recent literature on healthy subjects across different latitudes [13,38,39].

At the genus level, several taxa differed between sexes, with *Dorea*, *Roseburia*, and *Agathobacter* being more abundant in women than in men. All those genera [40] produce short-chain fatty acids (SCFAs) [41], which potentially offer protective effects against cardiovascular diseases. Several studies have found the same results [38,42,43].

It is worth noting that a bidirectional interaction between the host characteristics and microbiota composition seems to exist. Recent studies suggest that population-specific factors, including diet [39], age [44], and body composition [43,45,46], can change microbiome distribution across sexes at the phylum and genus levels. In the opposite direction, the gut microbiome has been implicated in cardiovascular disease and risk factors in a sex-dependent manner [19,20,45,47]. We focused our analysis on subjects with arterial stiffness.

Our study showed no significant differences in alpha or beta diversity measures between subjects with and without arterial stiffness for either sex. We found only one study that performed this analysis in a population with vascular stiffness (3087 patients from Potsdam, Germany), and no differences were found in diversity between sexes [18].

When considering only patients with arterial stiffness, we observed that the difference between the sexes in *Agathobacter*, *Dorea*, *Roseburia*, and *Blautia* abundances was amplified (increased log ratios), making the last one significant (also an SCFA producer). As all these bacteria contribute to cardiovascular health [38,42,43], we hypothesize that they can protect women, even if they have the same baseline risk when considering arterial stiffness measures. As we know, men have more cardiovascular events [16]. Studying these patients over time could clarify this aspect.

Roseburia has an inverse relationship with arterial stiffness in men. A previous study found that a lower abundance of *Roseburia* was associated with increased arterial stiffness in women [15]. There is also evidence that low levels of *Roseburia* are associated with higher blood pressure [48] and systemic atherosclerosis [49,50,51]. In our population of men, there was a relationship between the increase in the *Bifidobacterium* and *Subdoligranulum* genera and arterial stiffness. A previous study also linked *Subdoligranulum* to coronary heart disease in men [20]. This taxon is considered beneficial [52], yet its use as a probiotic has failed to show any beneficial effects in preclinical models [53]. As *Bifidobacterium* is a well-known probiotic, the results that link it to arterial stiffness are surprising. The *Bifidobacterium* genus comprises over 90 species, excluding the unclassified ones; not all benefit cardiovascular diseases [54,55]. Some studies suggested that the abundance of Bifidobacterium and its diversity and beneficial effects decline with age and several diseases. However, whether *Bifidobacterium* abundance has any causal relationship to these conditions [55] remains unknown.

This study has several strengths, including a rigorous evaluation of arterial stiffness, careful consideration of confounding factors, and specific analysis. By using multiple measures (cf-PWV, CAVI, and ba-PWV), this study offers a comprehensive assessment of vascular health compared to studies that rely on a single measure. We selected the twelve most abundant genera in the sample, allowing us to control for outliers and characterize the leading players in the gut microbiota. Multiple regression models with varying adjustment levels demonstrate a comprehensive approach to managing potential confounders. The methodology employed enables the identification of sex-specific associations that might be overlooked in combined analyses and emphasizes the significance of sex as a biological variable in research.

This study has certain limitations that should be noted. The cross-sectional design restricts our ability to establish causation, and the sample size is relatively small when divided by sex. Likewise, it should be considered that the proportion of women was higher than that of men, BMI was over 27 in both studied groups, 12.2% were smokers, 40% were people with hypertension, 35% had dyslipidemia, and 15% had hypothyroidism. Additionally, there may be unmeasured confounding variables. We made extensive efforts to minimize the impact of unmeasured confounders by measuring and considering a comprehensive set of variables in our analyses. While the cross-sectional design provides valuable insights, it does limit causal inference; we only considered the most abundant phyla (*Firmicutes* and *Bacteroidota*) and the 12 most abundant genera in our analysis. We only considered differences when the effect size was large (log > 1). More extensive studies could confirm these findings and potentially reveal additional, smaller effect sizes.

This study identifies sex differences in gut microbiota and their association with arterial stiffness. Future research should investigate whether the favorable bacterial profile observed in women results in fewer cardiovascular events and how environmental and cultural factors contribute to these microbiota differences. Additionally, it is important to examine the differential effects of Roseburia, Bifidobacterium, and Subdoligranulum on arterial stiffness in men and women through metabolomic analyses. Integrating multi-omics data will further enhance our understanding of the host–microbiome relationship.

This study paves the way for developing sex-specific biomarkers for cardiovascular risk assessment, creating personalized therapeutic approaches based on microbiota profiles, and investigating the potential for microbiota-based preventive strategies.

## 5. Conclusions

The women in the Spanish population under study had higher microbiome diversity and potentially protective genera. Likewise, this study found significant sex-specific differences in gut microbiota composition and association with arterial stiffness. These findings emphasize the importance of sex-specific approaches in microbiome research and cardiovascular health interventions. While these results are promising, larger multicenter studies with diverse populations are needed to validate these findings. Future research with expanded cohorts could clarify the links between sex-specific microbiota profiles and cardiovascular outcomes, enabling personalized prevention and treatment strategies.

## Figures and Tables

**Figure 1 nutrients-17-00053-f001:**
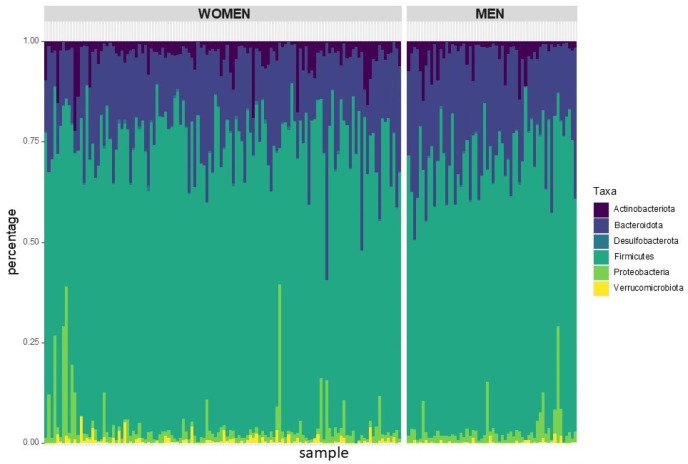
Phylum relative abundance by sex. *Firmicutes*: 70% (males (68%), female (71%); *p* < 0.01); *Bacteriodota*: 22% (males (25%), females (20%); *p* < 0.01).

**Figure 2 nutrients-17-00053-f002:**
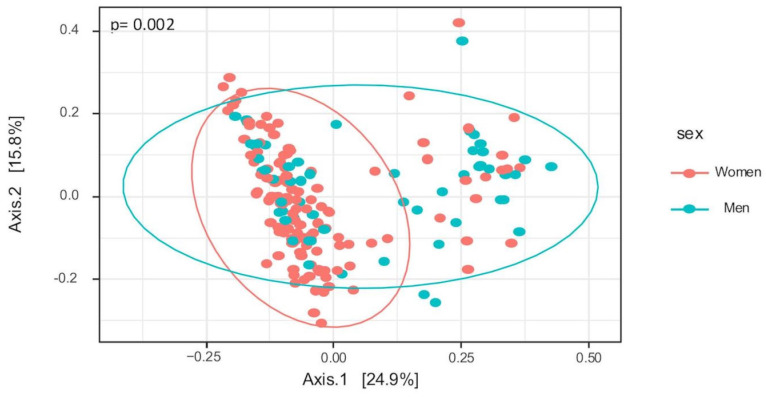
Principal Coordinate Analysis using Bray–Curtis Dissimilarity between sex (genus level). In this PCA plot, red points correspond to samples from women, and blue points correspond to male samples. Ellipses indicate group samples.

**Figure 3 nutrients-17-00053-f003:**
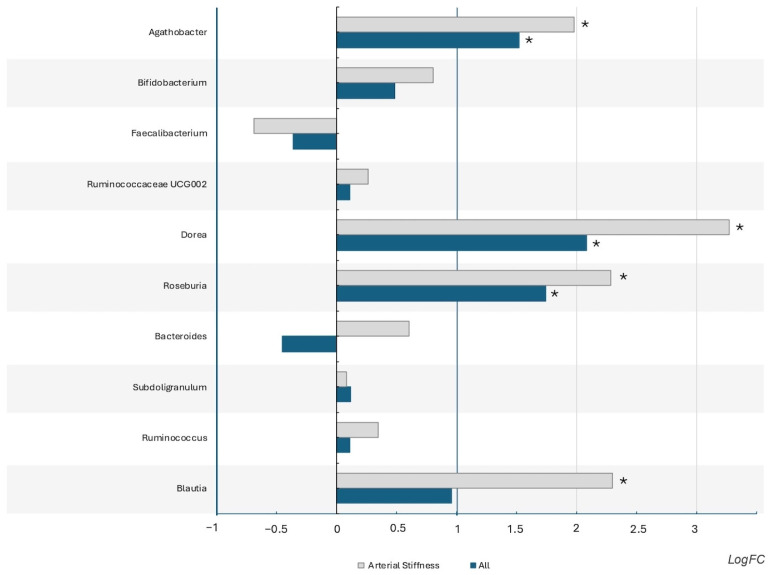
The bar chart shows differentially expressed genes in women vs. men, considering all the samples (blue) and only subjects with arterial stiffness (grey). FDR—false discovery rate; LogFC—log fold change. * significant results (*p* < 0.05; FDR < 0.05; log FC > −1 or >1).

**Figure 4 nutrients-17-00053-f004:**
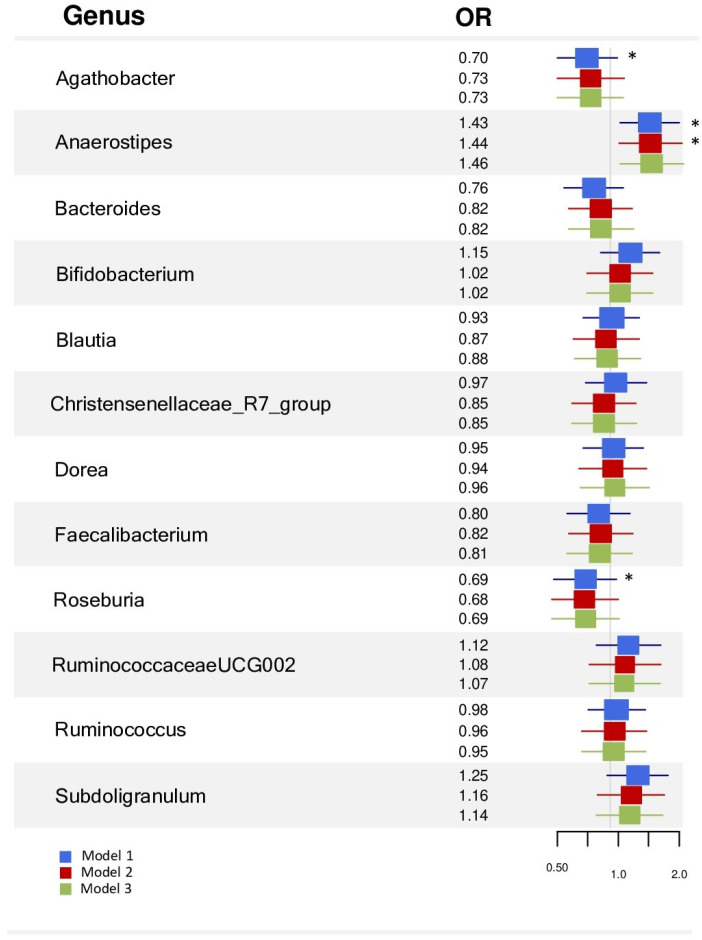
Forest plot showing the logistic regression results of microbiome abundance (independent variable) and arterial stiffness (dependent variable). Model 1: adjusted for sex and age. Model 2: adjusted for Model 1 and body mass index, hemoglobin A1c, total cholesterol, HDL cholesterol, LDL cholesterol, and systolic blood pressure. Model 3: adjusted for Model 2 and Mediterranean diet score. * *p*-value < 0.05.

**Figure 5 nutrients-17-00053-f005:**
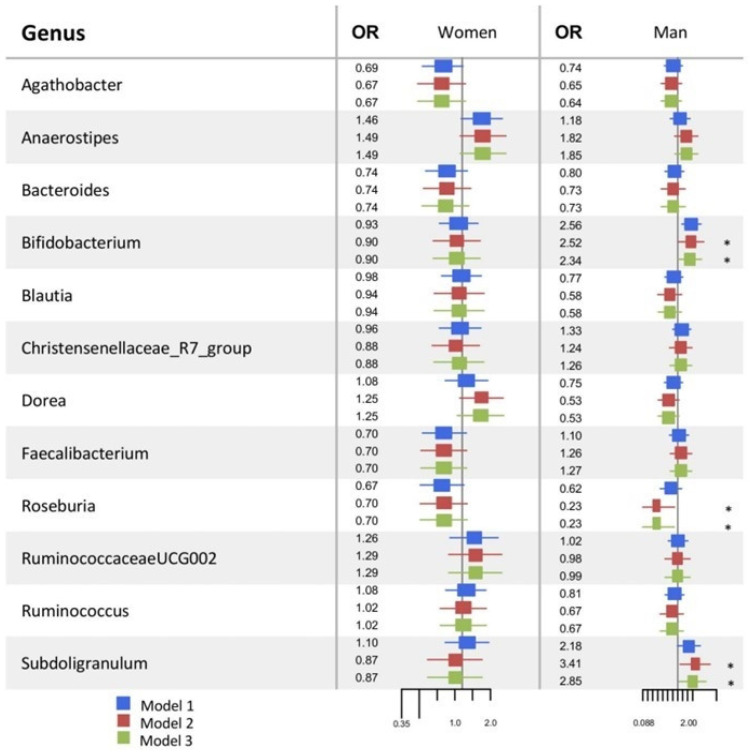
The forest plot shows the logistic regression results of microbiome abundance (independent variable) and arterial stiffness (dependent variable) in men and women. Model 1: adjusted for age. Model 2: adjusted for Model 1 and body mass index, hemoglobin A1c, total cholesterol, HDL cholesterol, LDL cholesterol, and systolic blood pressure. Model 3: adjusted for Model 2 and Mediterranean diet score. * *p*-value < 0.05.

**Table 1 nutrients-17-00053-t001:** Demographic and clinical characteristics by group.

	Total (180)		Women (n = 122; 68%)		Men (n = 58; 32%)		
	Mean/n	SD/%	Mean/n	SD/%	Mean/n	SD/%	*p*-Value
Age	62.6	7.5	63.1	7.3	61.6	7.9	0.19
Smoking							0.18
No smoker	91	50.6	63	51.6	28	48.3	
Current smoker	22	12.2	18	14.8	4	6.9	
Ex-smoker	67	37.2	41	33.6	26	44.8	
Body mass index (kg/m^2^)	27.1	4.6	27.2	5.2	27.0	3.2	0.80
Score MD	9.03	1.85	9.1	1.6	9.0	2.3	0.79
Adherence MD (%)	115	63.9	79	64.8	36	62.1	0.85
SBP (mmHg)	124	18	122.1	17.5	129.8	16.8	<0.01
DBP (mmHg)	79	10	77.8	9.9	82.3	8.6	<0.01
Heart rate (bpm)	67	10	69.4	9.8	63.0	9.7	<0.01
Triglycerides (mg/dL)	106	46	104.4	46.7	110.0	46.2	0.44
Total cholesterol (mg/dL)	196	30	199.8	29.0	186.6	29.2	<0.01
HDL cholesterol (mg/dL)	60	14	63.5	14.0	53.1	11.5	<0.01
LDL cholesterol (mg/dL)	116	27	117.4	27.7	114.1	25.8	0.47
Fasting blood glucose	91	12	90.8	12.5	92.7	11.2	0.35
HbA1c (%)	5.64	0.31	5.58	0.32	5.62	0.31	0.24
Creatinine (mg/dL)	0.81	0.16	0.74	0.12	0.95	0.16	<0.01
CKD-EPI (mL/min/1.72 m^2^)	85.4	13.2	85.4	13.1	85.4	13.7	10.98
Obesity	37	20.6	27	22.1	10	17.2	0.58
Hypertension	73	40.3	42	34.1	31	53.4	0.01
Dyslipidemia	63	35.4	48	39.4	15	25.9	0.07
Hypothyroidism	27	15.0	19	19.2	8	9.9	0.82
Antihypertensive drugs	55	30.4	32	26.0	23	39.7	0.06
Lipid-lowering drugs	59	33.1	45	36.9	14	24.1	0.08

Legend: LDL—low-density lipoprotein; HDL—high-density lipoprotein; MD—Mediterranean diet; SBP—systolic blood pressure; DBP—diastolic blood pressure; CKD-EPI—glomerular filtration rate by Chronic Kidney Disease Epidemiology Collaboration.

## Data Availability

The data supporting the findings of this study are available on ZENODO under https://zenodo.org/records/10407428 (accessed on 26 December 2024).

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
