# Peer review of "Sex Differences in Gut Microbiota and Their Relation to Arterial Stiffness (MIVAS Study)"

_nutrients, 2024, doi:10.3390/nu17010053_

Round 1

Reviewer 1 Report

Comments and Suggestions for Authors

The Authors conducted a very interesting study on the differences in microbiota diversity between men and women. Arterial stiffness was identified as an independent predictor of cardiovascular events and mortality, regardless of traditional risk factors.

The results are very interesting and indicate the direction of future research, which could also take into account other factors influencing the composition of the microbiome in this group of patients. 

It would be good to provide the frequency of cardiac events in women and men in the Spanish population and how this relates to the study results.

Minor corrections in references recommended according to instruction for authors.

Author Response

Dear Reviewer,

I would like to express our sincere gratitude for your thorough review of our manuscript. Your detailed comments and constructive suggestions have significantly improved the quality of our work. Your expertise and time spent analyzing our research are greatly appreciated.We trust that the changes we have made meet your expectations and address the points you raised.

Reviewer 1

The Authors conducted a very interesting study on the differences in microbiota diversity between men and women. Arterial stiffness was identified as an independent predictor of cardiovascular events and mortality, regardless of traditional risk factors.

Comments 1: The results are very interesting and indicate the direction of future research, which could also take into account other factors influencing the composition of the microbiome in this group of patients. 

Response 1: We have changed discussion ( page12, lines 601-608) of the discussion to include future directions.

This study identifies sex differences in gut microbiota and their association with arterial stiffness. Future research should investigate whether the favorable bacterial profile observed in women results in fewer cardiovascular events and how environmental and cultural factors contribute to these microbiota differences. Additionally, it is important to examine the differential effects of Roseburia, Bifidobacterium, and Subdoligranulum on arterial stiffness in men and women through metabolomic analyses. Integrating multi-omics data will further enhance our understanding of the host-microbiome relationship.

Comments 2: It would be good to provide the frequency of cardiac events in women and men in the Spanish population and how this relates to the study results.

Response 2: We added this information in the first paragraph of introduction (lines 59-62)

 In Spain, data from 2022 indicated that circulatory diseases, including coronary disease, cerebrovascular disease, cardiac insufficiency, and hypertensive disease, accounted for 267.87 deaths per 100,000 men and 174.12 deaths per 100,000 women.

Comments 3: Minor corrections in references are recommended according to instruction for authors.

Response 3: We reviewed the references following the MDPI style as outlined in the instructions for authors.

Reviewer 2 Report

Comments and Suggestions for Authors

Dear authors,

First of all, congratulations on the in-depth research you have conducted. Below, I will ask a few questions that you need to answer in order to help the reader better understand the message you are conveying.
The abstract is quite clearly written; I do not recommend any changes!

In the introduction, it should be more extensive and describe the local microbiota in women and men, and whether it is influenced by the Mediterranean diet. Does the consumption of processed or preserved foods modify the microbiota of patients?

In short, there is significant evidence that the Mediterranean diet positively influences the microbiota by promoting the growth of beneficial bacteria and increasing microbial diversity. On the other hand, the consumption of processed or preserved foods may have negative effects, potentially reducing microbiota diversity and promoting harmful bacterial populations. Additionally, the composition of the microbiota can vary between men and women, influenced by hormonal and lifestyle factors.
The subjects included in the study come from different environments, with different eating habits and practices. (Abrignani V, Salvo A, Pacinella G, Tuttolomondo A. The Mediterranean Diet, Its Microbiome Connections, and Cardiovascular Health: A Narrative Review. Int J Mol Sci. 2024 Apr 30;25(9):4942. doi: 10.3390/ijms25094942. PMID: 38732161; PMCID: PMC11084172)"

A limitation of the study is the significant difference between men and women, which should be highlighted as a study limitation.

Obesity is a well-known risk factor for cardiovascular diseases. A Body Mass Index (BMI) over 27 generally classifies individuals as overweight or obese. In clinical studies, patients with higher BMI values (especially over 30) are more likely to have associated comorbidities such as hypertension, type 2 diabetes, and other metabolic disorders, all of which increase cardiovascular risk.
This also represents another limitation of the study, considering that the BMI is over 27 in both studied groups. (Powell-Wiley TM, Poirier P, Burke LE, Després JP, Gordon-Larsen P, Lavie CJ, Lear SA, Ndumele CE, Neeland IJ, Sanders P, St-Onge MP; American Heart Association Council on Lifestyle and Cardiometabolic Health; Council on Cardiovascular and Stroke Nursing; Council on Clinical Cardiology; Council on Epidemiology and Prevention; and Stroke Council. Obesity and Cardiovascular Disease: A Scientific Statement From the American Heart Association. Circulation. 2021 May 25;143(21):e984-e1010. doi: 10.1161/CIR.0000000000000973. Epub 2021 Apr 22. PMID: 33882682; PMCID: PMC8493650)

Smoking is a well-known risk factor for cardiovascular diseases, as it contributes to the development of conditions such as hypertension, heart disease, and stroke. The statement suggests that including smokers or former smokers in the study interferes with the results, as smoking directly impacts cardiovascular health. It introduces an additional risk factor that may distort the study results.
In both groups, smokers or former smokers were included, which goes against the inclusion criteria (healthy patients). These should be listed as limitations of the study.
(https://www.cdc.gov/heart-disease/about/index.html; Gallucci G, Tartarone A, Lerose R, Lalinga AV, Capobianco AM. Cardiovascular risk of smoking and benefits of smoking cessation. J Thorac Dis. 2020 Jul;12(7):3866-3876. doi: 10.21037/jtd.2020.02.47. PMID: 32802468; PMCID: PMC7399440)

Cholesterol values are also at the risk threshold, which could lead to erroneous measurements!
HDL cholesterol is also in the risk zone for the male group. All these data can lead to erroneous study results.
Furthermore, the presence of hypertension (HTA), hypothyroidism, and dyslipidemia represents limitations, even if you have set other inclusion criteria.

Regarding the evaluation of vascular stiffness (cf-PWV, CAVI, ba-PWV), some clarifications are needed—was it performed by one person or multiple people? This is important to eliminate variations in interpretation.This should be specified.
The study limitations need to be completed.

The conclusions should suggest the need for further studies to extend the research to larger patient groups

Author Response

Dear Reviewer,

I would like to express our sincere gratitude for your thorough review of our manuscript. Your detailed comments and constructive suggestions have significantly improved the quality of our work. Your expertise and time spent analyzing our research are greatly appreciated.We trust that the changes we have made meet your expectations and address the points you raised.

Reviewer 2

First of all, congratulations on the in-depth research you have conducted. Below, I will ask a few questions that you need to answer in order to help the reader better understand the message you are conveying.

The abstract is quite clearly written; I do not recommend any changes!

Comment 1: In the introduction, it should be more extensive and describe the local microbiota in women and men, and whether it is influenced by the Mediterranean diet. Does the consumption of processed or preserved foods modify the microbiota of patients?.

In short, there is significant evidence that the Mediterranean diet positively influences the microbiota by promoting the growth of beneficial bacteria and increasing microbial diversity. On the other hand, the consumption of processed or preserved foods may have negative effects, potentially reducing microbiota diversity and promoting harmful bacterial populations. Additionally, the composition of the microbiota can vary between men and women, influenced by hormonal and lifestyle factors.    The subjects included in the study come from different environments, with different eating habits and practices. (Abrignani V, Salvo A, Pacinella G, Tuttolomondo A. The Mediterranean Diet, Its Microbiome Connections, and Cardiovascular Health: A Narrative Review. Int J Mol Sci. 2024 Apr 30;25(9):4942. doi: 10.3390/ijms25094942. PMID: 38732161; PMCID: PMC11084172)".

Response1: We sincerely appreciate this comment, as it has prompted us to reevaluate the information we have available thus far.

We would like to emphasize that while there exists an article that describe the Spanish population and its relationship to the Mediterranean diet, it is important to note that of the 530 individuals mentioned in that study, only 5% (28 patients) were from Castilla-León, the region where Salamanca is located.

Additionally, we chose to focus on a population of both men and women living in the same city, which we believe helps to minimize cultural disparities. Furthermore, both genders exhibit a rate of adherence to the Mediterranean diet of approximately 60%.

That being said, we’re excited to enhance our introduction.

  1. We added a sentence describind the findings of the study performed in the Spanish population. (Introduction, page 2, 83-88).

In this study involving 530 Spanish patients, the gut microbiome was primarily characterized by Firmicutes (~53.9%) and Bacteroidota (~37.2%). The most prevalent genera included Bacteroides (~18.4%) and Faecalibacterium (~12.5%), followed by Prevotella (6.7%), Alistipes (~3.4%), and Oscillospiraceae (~2.3%). Gut microbiota was rich (Chao 283 ) and diverse (Shannon 4.3). The levels of the phylum Proteobacteria and the genus Faecalibacterium were significantly greater in males compared to females.

  1. We also added two paragraphs to explain the relatioship of arterial stiffnes wtih cardiovascular risk factors, gut microbiota and lifestyle, including your suggestions on diet behaviours. We hope this stayed as expected. ( page 2, line 53-80.

Recent efforts have concentrated on enhancing the identification of individuals at high risk for cardiovascular events by employing biomarkers that signal early changes before overt disease manifests [3]. Arterial stiffness has proven to be a reliable predictor of cardiovascular disease across diverse populations [4], including the general population, hypertensive individuals, the elderly, and patients with type 2 diabetes or end-stage renal disease [5]. The main determinants of arterial stiffness encompass major cardiovascular risk factors [1], such as age, hypertension, gender, obesity, smoking, dyslipidemia, hypothyroidism, and influences from genetic predisposition, systemic inflammation, and gut microbiota composition.

The transition from a healthy microbiota to dysbiosis involves increased pathobionts and is often triggered by environmental factors. [6] While age and genetic composition play significant roles, diet and lifestyle are key contributors to microbiome variation. The Mediterranean diet enhances microbial diversity and promotes beneficial bacteria[7], whereas artificial sweeteners and dietary emulsifiers can disrupt gut microbiota and cause inflammation [8]. Variations also exist between omnivores and vegans [9] and those with high protein versus high carbohydrate intake. [10] Obesity is linked to reduced fecal microbial diversity [11]. Antibiotic use can cause rapid declines in microbial diversity. [12]

Comment 2:

Obesity is a well-known risk factor for cardiovascular diseases. A Body Mass Index (BMI) over 27 generally classifies individuals as overweight or obese. In clinical studies, patients with higher BMI values (especially over 30) are more likely to have associated comorbidities such as hypertension, type 2 diabetes, and other metabolic disorders, all of which increase cardiovascular risk. This also represents another limitation of the study, considering that the BMI is over 27 in both studied groups.    (Powell-Wiley TM, Poirier P, Burke LE, Després JP, Gordon-Larsen P, Lavie CJ, Lear SA, Ndumele CE, Neeland IJ, Sanders P, St-Onge MP; American Heart Association Council on Lifestyle and Cardiometabolic Health; Council on Cardiovascular and Stroke Nursing; Council on Clinical Cardiology; Council on Epidemiology and Prevention; and Stroke Council. Obesity and Cardiovascular Disease: A Scientific Statement From the American Heart Association. Circulation. 2021 May 25;143(21):e984-e1010. doi: 10.1161/CIR.0000000000000973. Epub 2021 Apr 22. PMID: 33882682; PMCID: PMC8493650)

Smoking is a well-known risk factor for cardiovascular diseases, as it contributes to the development of conditions such as hypertension, heart disease, and stroke. The statement suggests that including smokers or former smokers in the study interferes with the results, as smoking directly impacts cardiovascular health. It introduces an additional risk factor that may distort the study results. In both groups, smokers or former smokers were included, which goes against the inclusion criteria (healthy patients). These should be listed as limitations of the study.
(https://www.cdc.gov/heart-disease/about/index.html; Gallucci G, Tartarone A, Lerose R, Lalinga AV, Capobianco AM. Cardiovascular risk of smoking and benefits of smoking cessation. J Thorac Dis. 2020 Jul;12(7):3866-3876. doi: 10.21037/jtd.2020.02.47. PMID: 32802468; PMCID: PMC7399440)

Cholesterol values are also at the risk threshold, which could lead to erroneous measurements!
HDL cholesterol is also in the risk zone for the male group. All these data can lead to erroneous study results.

Furthermore, the presence of hypertension (HTA), hypothyroidism, and dyslipidemia represents limitations, even if you have set other inclusion criteria.

Response 2: Thank you for pointing this out. We agree with the comment and understood that we made a mistake when defining groups. Of course we don´t have a healthy group as we include patients with cardiovascular risk factors in our population.

After careful consideration, we changed the group's name from healthy to without arterial stiffness to refer to those who did not meet the criteria for arterial stiffness.

This led to a diverse number of changes throughout the article:

Abstract. Page 1, line 41: All other cases were considered subjects without arterial stiffness.

Methods. Page 5, line 218:: All other cases were considered subjects without arterial stiffness.

Methods. Page 5, line 237: as the dependent variable having or not arterial stiffness.

Results. Page 7, lines 290-291:  between subjects with and without arterial stiffness.

Discussion. Page 11, line 356:  between subjects with and without arterial stiffness for either sex

Supplementary Material, Page 1, flow chart:  subjects without arterial stiffness

We also added two paragraphs to explain the relatioship of arterial stiffnes wtih cardiovascular risk factors, gut microbiota and lifestyle, including your suggestions on diet behaviours. We hope this stayed as expected, page 2, line 53-80:

Recent efforts have concentrated on enhancing the identification of individuals at high risk for cardiovascular events by employing biomarkers that signal early changes before overt disease manifests [3]. Arterial stiffness has proven to be a reliable predictor of cardiovascular disease across diverse populations [4], including the general population, hypertensive individuals, the elderly, and patients with type 2 diabetes or end-stage renal disease [5]. The main determinants of arterial stiffness encompass major cardiovascular risk factors [1], such as age, hypertension, gender, obesity, smoking, dyslipidemia, hypothyroidism, and influences from genetic predisposition, systemic inflammation, and gut microbiota composition.

The transition from a healthy microbiota to dysbiosis involves increased pathobionts and is often triggered by environmental factors. [6] While age and genetic composition play significant roles, diet and lifestyle are key contributors to microbiome variation. The Mediterranean diet enhances microbial diversity and promotes beneficial bacteria[7], whereas artificial sweeteners and dietary emulsifiers can disrupt gut microbiota and cause inflammation [8]. Variations also exist between omnivores and vegans [9] and those with high protein versus high carbohydrate intake. [10] Obesity is linked to reduced fecal microbial diversity [11]. Antibiotic use can cause rapid declines in microbial diversity. [12]

In addition, we have included the following sentence in the discussion section, page 11, lines 374-376:

Likewise, it should be considered that the proportion of women is higher than men, the BMI is over 27 in both studied groups, the 12.2% were smokers, 40% were people with hypertension, 35% with dyslipidemia and 15% with hypothyroidism.

Comment 3: Regarding the evaluation of vascular stiffness (cf-PWV, CAVI, ba-PWV), some clarifications are needed—was it performed by one person or multiple people? This is important to eliminate variations in interpretation. This should be specified.

Response 3: On methods section, page 4, section 2.3,line 153-154, we specify the number of evaluators, and the standardized criteria followed to avoid variability, being as follows:

A standardized protocol was used to train two research nurses to take measurements and answer questionnaires before the study began. [11]

Comment 4: The study limitations need to be completed.

Response 4:

We have included the following sentence in the discussion section, page 11, lines 374-376:

Likewise, it should be considered that the proportion of women is higher than men, the BMI is over 27 in both studied groups, the 12.2% were smokers, 40% were people with hypertension, 35% with dyslipidemia and 15% with hypothyroidism.

Comment 5: The conclusions should suggest the need for further studies to extend the research to larger patient groups

Response 5: Accordingly, with this recommendation, we added two more sentences to conclusion section, page 12, lines 400-403:

While these results are promising, larger multicenter studies with diverse populations are needed to validate these findings. Future research with expanded cohorts could clarify the links between sex-specific microbiota profiles and cardiovascular outcomes, enabling personalized prevention and treatment strategies.

Reviewer 3 Report

Comments and Suggestions for Authors

Dear Authors,

Below, I have provided my comments and suggestions to further enhance the quality of the manuscript:

ABSTRACT

  • This section could be better structured by including a brief background before presenting the study's objectives.
  • The ethical considerations of the study, currently placed at the end of the abstract, should be relocated to the Methods section and the appropriate section at the end of the manuscript.

INTRODUCTION

  • This section could benefit from further elaboration. In the first part, I would expand the description of cardiovascular diseases (CVD), providing more details and incorporating epidemiological data.
  • Following that, I would provide a clearer connection between current studies, linking them more effectively using transition sentences.

METHODS

  • This section is adequately presented. However, I recommend clearly defining the reporting guidelines followed, according to EQUATOR NETWORK, in the initial part of the section. This would significantly enhance the methodological rigor of the study.

RESULTS AND DISCUSSION

  • In Table 1, the list of acronyms should be included in the "legend" section rather than as part of the table itself.
  • The quality of Figures 3, 4, and 5 is quite poor; I kindly ask for them to be revised as they are difficult to read.
    Both of these sections are well-written with high methodological quality and a smooth, readable text. However, I would suggest expanding the discussion at the end to include the future implications of the study.

CONCLUSIONS
This section should be revised and expanded.

General Remarks
This manuscript is of adequate quality, and in my opinion, after the suggested revisions, it could be considered for publication. I congratulate the authors on their work.

Author Response

Dear Reviewer,

I would like to express our sincere gratitude for your thorough review of our manuscript. Your detailed comments and constructive suggestions have significantly improved the quality of our work. Your expertise and time spent analyzing our research are greatly appreciated.We trust that the changes we have made meet your expectations and address the points you raised.

ABSTRACT

Comment 1: This section could be better structured by including a brief background before presenting the study's objectives.

Response 1: We have added the following paragraph before the objective of the abstract, page 1, line 33-35:

Recent research highlights the potential role of sex-specific variations in cardiovascular disease. The gut microbiome has been shown to differ between sexes in patients with cardiovascular risk factors.

Comment 2: The ethical considerations of the study, currently placed at the end of the abstract, should be relocated to the Methods section and the appropriate section at the end of the manuscript.

Response 2: Ethical considerations can be found at the end of the method, section on page 6, line 256-262, and in the corresponding section at the end of the manuscript, page 12, line 451-451.

INTRODUCTION

Comment 3: This section could benefit from further elaboration. In the first part, I would expand the description of cardiovascular diseases (CVD), providing more details and incorporating epidemiological data.

Response 3: We incorporated the following sentence in the first paragraph of introduction, page 2, lines 59-62:

 In Spain, data from 2022 indicated that circulatory diseases, including coronary disease, cerebrovascular disease, cardiac insufficiency, and hypertensive disease, accounted for 267.87 deaths per 100,000 men and 174.12 deaths per 100,000 women.

Comment 4: Following that, I would provide a clearer connection between current studies, linking them more effectively using transition sentences.

Response 4: We did a extensive review of introduction , we hope that now it meets your expectations, so that first 4 paragraphs stayed as below, (pag 2, 57-98)

Cardiovascular disease (CVD) remains a leading cause of morbidity and mortality worldwide[1], and growing evidence suggests significant sex differences in its prevalence, progression, and outcomes. In Spain, data from 2022 indicated that circulatory diseases, including coronary disease, cerebrovascular disease, cardiac insufficiency, and hypertensive disease, accounted for 267.87 deaths per 100,000 men and 174.12 deaths per 100,000 women[2].

Recent efforts have concentrated on enhancing the identification of individuals at high risk for cardiovascular events by employing biomarkers that signal early changes before overt disease manifests [3]. Arterial stiffness has proven to be a reliable predictor of cardiovascular disease across diverse populations [4], including the general population, hypertensive individuals, the elderly, and patients with type 2 diabetes or end-stage renal disease [5]. The main determinants of arterial stiffness encompass major cardiovascular risk factors [1], such as age, hypertension, gender, obesity, smoking, dyslipidemia, hypothyroidism, and influences from genetic predisposition, systemic inflammation, and gut microbiota composition.

The transition from a healthy microbiota to dysbiosis involves increased pathobionts and is often triggered by environmental factors. [6] While age and genetic composition play significant roles, diet and lifestyle are key contributors to microbiome variation. The Mediterranean diet enhances microbial diversity and promotes beneficial bacteria[7], whereas artificial sweeteners and dietary emulsifiers can disrupt gut microbiota and cause inflammation [8]. Variations also exist between omnivores and vegans [9] and between those with high protein versus high carbohydrate intake. [10] Obesity is linked to reduced fecal microbial diversity [11]. Antibiotic use can cause rapid declines in microbial diversity. [12]

The gut microbiome is also being studied by its difference between males and females[13], and these differences have been recently documented in the Spanish population [14]. In this study involving 530 Spanish patients, the gut microbiome was primarily characterized by Firmicutes (~53.9%) and Bacteroidota (~37.2%). The most prevalent genera included Bacteroides (~18.4%) and Faecalibacterium (~12.5%), followed by Prevotella (6.7%), Alistipes (~3.4%), and Oscillospiraceae (~2.3%). Gut microbiota was rich (Chao 283 ) and diverse (Shannon 4.3). The levels of the phylum Proteobacteria and the genus Faecalibacterium were significantly greater in males compared to females.

METHODS

Comment 5: This section is adequately presented. However, I recommend clearly defining the reporting guidelines followed, according to EQUATOR NETWORK, in the initial part of the section. This would significantly enhance the methodological rigor of the study.

Response 5: The strobe checklist is included in supplementary material and the following sentence is added in methods, and design section (page 3, line 124).

The Strobe checklist is included in supplementary material.

RESULTS AND DISCUSSION

Comment 6: In Table 1, the list of acronyms should be included in the "legend" section rather than as part of the table itself.

Response 6: We removed the list of acronyms from the table and relocated it to the "legend" below the table, as this specific journal does not have a dedicated section for figure legends at submission. We added two missing acronyms and removed one that was a typo.

Legend: LDL, low-density lipoprotein, HDL,high-density lipoprotein. MD: Mediterranean diet; SBP: Systolic blood pressure; DBP: Diastolic blood pressure; CKD-EPI: glomerular filtration rate by Chronic Kidney Disease Epidemiology Collaboration.

Comment 7: The quality of Figures 3, 4, and 5 is quite poor; I kindly ask for them to be revised as they are difficult to read.

Response 7:  We replaced the figures for others of higher quality.

Comment 8: Both of these sections are well-written with high methodological quality and a smooth, readable text. However, I would suggest expanding the discussion at the end to include the future implications of the study.

Response 8: In response to this request we reviewed the discussion and replaced the two last paragraphs, page 12, 603-612):

This study identifies sex differences in gut microbiota and their association with arterial stiffness. Future research should investigate whether the favorable bacterial profile observed in women results in fewer cardiovascular events and how environmental and cultural factors contribute to these microbiota differences. Additionally, it is important to examine the differential effects of Roseburia, Bifidobacterium, and Subdoligranulum on arterial stiffness in men and women through metabolomic analyses. Integrating multi-omics data will further enhance our understanding of the host-microbiome relationship.

This study paves the way to develop sex-specific biomarkers for cardiovascular risk assessment, create personalized therapeutic approaches based on microbiota profiles and investigate the potential for microbiota-based preventive strategies.

CONCLUSIONS

Comment 9: This section should be revised and expanded.

Response 9: Accordingly, with this recommendation, we added two more sentences to conclusion section, page 12, lines 400-403:

While these results are promising, larger multicenter studies with diverse populations are needed to validate these findings. Future research with expanded cohorts could clarify the links between sex-specific microbiota profiles and cardiovascular outcomes, enabling personalized prevention and treatment strategies.

General Remarks
This manuscript is of adequate quality, and in my opinion, after the suggested revisions, it could be considered for publication. I congratulate the authors on their work.

Round 2

Reviewer 3 Report

Comments and Suggestions for Authors

The authors have made appropriate changes to the manuscript. It may be published